# Fungistatic Action of N-Acetylcysteine on *Candida albicans* Biofilms and Its Interaction with Antifungal Agents

**DOI:** 10.3390/microorganisms8070980

**Published:** 2020-06-30

**Authors:** Thaís Soares Bezerra Santos Nunes, Leticia Matheus Rosa, Yuliana Vega-Chacón, Ewerton Garcia de Oliveira Mima

**Affiliations:** Laboratory of Applied Microbiology, Department of Dental Materials and Prosthodontics, School of Dentistry, São Paulo State University (UNESP), Araraquara 14801-903, SP, Brazil; thais.soaresbsn@gmail.com (T.S.B.S.N.); lemrosa@hotmail.com (L.M.R.); yuliana.v.chacon@gmail.com (Y.V.-C.)

**Keywords:** acetylcysteine, biofilms, extracellular matrix, *Candida albicans*, drug resistance, antifungal agents

## Abstract

Therapies targeted to fungal biofilms, mainly against the matrix, and therapies that do not induce microbial resistance are relevant. N-acetylcysteine (NAC), a mucolytic agent, has shown antimicrobial action. This study evaluated the effect of NAC against fluconazole-susceptible (CaS) and -resistant (CaR) *Candida albicans*. The susceptibility of planktonic cultures to NAC, the effect of NAC on biofilms and their matrix, the interaction of NAC with antifungal agents, and confocal microscopy were evaluated. Data were analyzed descriptively and by the ANOVA/Welch and Tukey/Gomes–Howell tests. The minimum inhibitory concentration (MIC) of NAC was 25 mg/mL for both strains. NAC significantly reduced the viability of both fungal strains. Concentrations higher than the MIC (100 and 50 mg/mL) reduced the viability and the biomass. NAC at 12.5 mg/mL increased the fungal viability. NAC also reduced the soluble components of the biofilm matrix, and showed synergism with caspofungin against planktonic cultures of CaS, but not against biofilms. Confocal images demonstrated that NAC reduced the biofilm thickness and the fluorescence intensity of most fluorochromes used. High concentrations of NAC had similar fungistatic effects against both strains, while a low concentration showed the opposite result. The antibiofilm action of NAC was due to its fungistatic action.

## 1. Introduction

*Candida albicans* is an opportunistic fungal pathogen found on human mucosae, such as those of the oral, gastrointestinal, and genitourinary tracts [1]. *C. albicans* exhibits the ability to adhere to surfaces and form biofilms, which are complex communities of microorganisms attached to biotic or abiotic surfaces and surrounded by a self-produced matrix [2]. Under specific host conditions, especially immunodeficiency, *C. albicans* causes a local infection known as candidiasis, which may spread systemically and result in candidemia, a nosocomial infection with high mortality rates [3]. Furthermore, *C. albicans* possess the ability to undergo morphological transition from commensal yeast to filamentous forms (pseudohyphae and hyphae), in a process known as polymorphism [4]. In the hyphal form, *C. albicans* is pathogenic, with the ability to invade host tissue [5]. Therefore, polymorphism is a significant virulence factor of this opportunistic fungal species [4,6].

*C. albicans* also grows as biofilms on biomedical devices, such as catheters and dental prostheses [7]. Oral candidiasis associated with denture use, known as denture stomatitis, is an oral infection prevalent in up to 70% of denture users, with larger proportions of females affected than males. Despite the multifactorial etiology of denture stomatitis, biofilm formation by *Candida* spp. on the denture surface is considered the main etiological factor [8]. Moreover, candidal biofilms formed in indwelling catheters act as a source of infection in candidemia, whose treatment requires both antifungal administration and timely removal of the catheter [9].

The biofilm matrix protects microbial cells growing within the biofilm from external agents, and acts as a barrier to antimicrobials and the host’s immune cells. Thus, in a biofilm, microbial cells exhibit higher tolerance to antimicrobials [10]. The matrix is also responsible for the three-dimensional architecture and mechanical properties of biofilms, such as their adhesive strength, cohesion, and stiffness, which enables co-aggregation of microorganisms and facilitates formation of microcolonies. The environment of biofilm modulates gene expression, metabolism, and cell signaling by molecules, influencing microbial behavior (quorum sensing) [11]. The biofilm matrix of *C. albicans* is composed of carbohydrates, such as β-glucans and mannan, as well as proteins, lipids, and extracellular DNA (eDNA) [12], and hexosamines, uronic acid, and phosphorus [13]. The *Candida albicans* biofilm matrix, especially the polysaccharides, contributes to co-aggregation of bacteria and protects bacterial cells against antibacterial agents [14,15,16,17]. *Candida albicans* biofilms also produce extracellular vesicles, whose cargo (polysaccharides and proteins) forms the biofilm matrix and confers drug tolerance [18].

Antifungal agents used to treat infections caused by *Candida* spp. are divided into the following three main classes: polyenes, azoles (imidazoles and triazoles), and echinocandins [19]. Nystatin is a topical polyene used to treat palatal mucosa affected by denture stomatitis [20,21]; fluconazole (triazole) and caspofungin (echinocandin) are used for the treatment of invasive candidiasis [9]. However, *C. albicans* cells growing within biofilms have been shown to exhibit 4 to 128 times lower antifungal susceptibility than their planktonic counterparts [22]. In addition to the inherent microbial tolerance of biofilms, the widespread use and misuse of antifungals have led to the development of antifungal resistance [6]. Such resistance leads to treatment failure with commercially available antifungal agents, and persistent infection. As a result, antimicrobial resistance remains a global threat to public health [23]. Therefore, the development of treatments aimed at overcoming antifungal resistance and disrupting biofilm matrix on medical devices is critical.

*N*-acetylcysteine (NAC) is a mucolytic and thiol compound used to reduce the viscosity of pulmonary mucous in patients with chronic respiratory diseases, such as bronchitis [24] and cystic fibrosis [25], as well as being used to treat paracetamol (acetominophen) overdose [26]. The sulfhydryl component of the NAC molecule breaks the disulfide bonds in mucous [27], and also acts as a precursor for glutathione synthesis and as a reactive oxygen species (ROS) scavenger, conferring an antioxidant effect to NAC [28]. Other studies have demonstrated antimicrobial and antibiofilm properties of NAC against pathogenic bacteria, such as *Mycobacterium tuberculosis* [29], *Klebsiella pneumoniae*, *Escherichia coli*, *Pseudomonas aeruginosa* [30,31], methicillin-resistant *Staphylococcus aureus* and *Staphylococcus epidermidis* [32], *Prevotella intermedia* [33], *Actinomyces naeslundii*, *Lactobacillus salivarius*, *Streptococcus mutans*, and *Enterococcus faecalis* [34,35]. Studies have also shown synergism between NAC and antibiotics [31,32,33,36,37]. There have also been reports that NAC disrupts mature bacterial biofilms and reduces the matrix polysaccharides [31,38]. In this context, investigating new applications of existing drugs (drug repurposing) has the advantage of reducing the time and cost spent in the development of new medications, such as by negating the need for clinical safety trials [39]. 

The antifungal and antibiofilm action of NAC alone or in association with antifungal agents (fluconazole, ketoconazole, and amphotericin B) against *C. albicans* has also been reported [37,40,41]. However, the antifungal effects of NAC against resistant strains and its effects on the components of the *C. albicans* biofilm matrix, to the best of our knowledge, are still unknown. Therefore, the aim of this study was to evaluate the effect of NAC on biofilms of fluconazole-resistant (CaR) and -susceptible *C. albicans* (CaS). Aside from the fungal viability and biomass, we focused on the effects of NAC on the components of the biofilm matrix: polysaccharides, proteins, lipids, and eDNA. We also evaluated the interaction of NAC with agents belonging to the following three classes of antifungals: polyenes (nystatin), azole (fluconazole), and echinocandin (caspofungin). Additionally, we analyzed biofilms treated with NAC using confocal scanning laser microscopy.

## 2. Materials and Methods 

### 2.1. Strains and Culture Conditions

Standard strains of CaS (SC5314) and CaR (ATCC 96901—American Type Culture Collection, Rockville, MD, USA) were stored at −80 °C in yeast nitrogen broth (YNB; Difco, Detroit, MI, USA) with glycerol 50%; they were thawed and reactivated in Sabouraud dextrose agar with 0.05 mg/mL of chloramphenicol (SDA; Acumedia Manufacturers Inc., Lansing, MI, USA) at 37 °C for 48 h. Five colonies of each strain were transferred to YNB with 100 mM of glucose (YNBg) and incubated overnight at 37 °C. Each fungal suspension was diluted 1:10 in fresh YNBg and incubated at 37 °C until the optical density at 540 nm (OD_540_) reached the mid-log phase of growth (mean ± standard deviation, SD, of OD_540_: 0.424 ± 0.042 and 0.460 ± 0.030 arbitrary units, au, for CaS and CaR, respectively), corresponding to 6.64 × 10^6^ ± 2.72 × 10^6^ and 6.32 × 10^6^ ± 1.49 × 10^6^ colony forming units per milliliter (CFU/mL) for CaS and CaR, respectively. Then, each fungal suspension was centrifuged (10,000× *g* for 10 min at 4 °C) and resuspended in Roswell Park Memorial Institute (RPMI) 1640 medium—bicarbonate free (Sigma-Aldrich, St. Louis, MO, USA, product code R6504-10X1L), buffered with morpholinepropanesulfonic acid (MOPS; Sigma-Aldrich), and supplemented with 2% D-glucose (Synth, São Paulo, Brazil), pH 7.0 (RPMId). Then, the OD_540_ for each strain was measured (0.585 ± 0.167 and 0.498 ± 0.005 au for CaS and CaR, respectively).

### 2.2. Antifungal Evaluation of NAC

NAC (Sigma-Aldrich) was diluted in RPMId at 100 mg/mL, according to the manufacturer’s instructions, for use in antifungal evaluation. The minimum inhibitory concentration (MIC) and minimum fungicidal concentration (MFC) were evaluated by the microdilution method, according to the recommendations of the Clinical and Laboratory Standards Institute (CLSI) [42] and the European Committee for Antimicrobial Susceptibility Testing (EUCAST) [43], with some modifications. Aliquots (100 µL) of twofold serial dilutions of NAC (final concentrations of 50 to 12.5 mg/mL) in RPMId were incubated with 100 µL of each fungal suspension individually for 24 h at 37 °C in 96-well, U-bottom microtiter plates (TPP, Trasadingen, Switzerland). A temperature of 37 °C was chosen, instead of 35 °C as recommended by the CLSI, for consistency with human body temperature and the temperature used for in vitro biofilm formation [44]. Aliquots (200 µL) of each fungal suspension with RPMId and of RPMId alone were used as controls (without NAC). For the test samples, each fungal suspension was diluted in RPMId with NAC to give a final concentration of 0.5–2.5 × 10^3^ CFU/mL. After incubation, the OD_540_ was measured in a spectrophotometer (FLUOstar Omega microplate reader; BMG Labtech GmbH, Offenburg, Germany), to obtain quantitative values and avoid any bias that might arise from visual inspection. The lowest concentration whose OD_540_ value was similar to that observed for RPMId alone was defined as the MIC. In addition, samples were plated on SDA and incubated at 37 °C for 48 h. After incubation, the colonies were counted (CFU/mL) and the lowest NAC concentration that inhibited colony growth was defined as the MFC.

### 2.3. Time-Kill Curves

Time-kill curves were performed using the same protocol as for the MIC analysis above. Both strains were grown until mid-log phase, as described in Section 2.1. Aliquots of 100 µL of NAC were added to each fungal suspension (100 µL) to achieve final NAC concentrations of 50 to 0.09 mg/mL for CaS, and of 50 to 12.5 mg/mL for CaR, using a 96-well, U-bottom microtiter plate. RPMId alone (without NAC) with the fungal inocula were used as negative controls. The final concentration of fungal suspension was 0.5–2.5 × 10^3^ CFU/mL. The absorbances of the samples in the plates were measured with a spectrophotometer (at 540 nm) during 24 h of incubation at 37 °C, with readings taken every 2 h.

### 2.4. Evaluation of NAC Activity on Biofilm Formation

Aliquots (200 µL) of each fungal suspension were transferred to 96-well, flat-bottom microtiter plates (Kasvi, São José dos Pinhais, Brazil) and incubated at 37 °C for 90 min (adhesion phase). Then, the medium from each well was aspirated, and each well was washed twice with 200 µL of sterile phosphate-buffered saline (PBS—0.136 M NaCl, 1 mM KH_2_PO_4_, 2 mM KCl, 10 mM Na_2_HPO_4_, pH 7.4) to remove non-adherent cells. NAC at 100, 50, 25, and 12.5 mg/mL in RPMId was added to each well (200 µL) and the plates incubated at 37 °C for 48 h to allow biofilm formation [44]. Control samples received only RPMId without NAC.

After 48 h, the biofilms were washed twice with 200 µL of PBS, and biofilm viability (CFU/mL) and total biomass (using the crystal violet assay) [45] were assessed. For viability analysis, 200 µL of PBS was added into each well, biofilms were individually scraped using a pipette tip, and tenfold serial dilutions in PBS were plated in duplicate on SDA, which was incubated at 37 °C for 48 h for colony counting. For biofilm total biomass analysis, 200 µL of 80% methanol was added to each well. After 15 min, the methanol was removed; then, 200 µL of 1% crystal violet was added to each well. After 5 min, samples were washed five times with PBS, and 200 µL of 33% acetic acid was added to remove the stain. This solution was transferred to another 96-well microtiter plate, and the absorbance at 570 nm was measured [45].

### 2.5. Effect of the Incubation Time with NAC on Biofilm Formation

The effect of NAC on biofilm formation was evaluated as described above, but the incubation period was reduced. The samples were evaluated for biofilm viability and total biomass after 6, 12, and 24 h of biofilm formation, with and without (control) NAC.

### 2.6. Evaluation of the Effect of NAC on Pre-Formed Biofilms 

Biofilms of each strain were grown as described above, without NAC, for 48 h. After the first 24 h of incubation, 100 µL of each well sample was removed and 100 µL of fresh RPMId was added. After 48 h, the biofilms were washed twice with PBS, and NAC at 100, 50, 25, and 12.5 mg/mL in RPMId was added to the wells (200 µL) and incubated at 37 °C for 24 h. Controls received only RPMId without NAC. After incubation, the biofilm viability (CFU/mL) and total biomass were evaluated as described above.

### 2.7. Effect of NAC on the Composition of the Biofilm Matrix

In order to evaluate the effect of NAC on the composition of the biofilm matrix, biofilms of each fungal strain were grown in triplicate in 24-well microtiter plates (Kasvi) using 2.5 mL of individual fungal suspension in each well. NAC was added after the adhesion phase (50 and 12.5 mg/mL for biofilm formation experiments) or after 48 h of biofilm growth (100 mg/mL for pre-formed biofilm experiments, followed by incubation for 24 h), as described above. Biofilms were washed twice with sterile 0.89% NaCl, and after the addition of 2.5 mL of 0.89% NaCl they were scraped using a pipette tip for 1 min. The content of triplicate wells (7.5 mL) was transferred to a single Falcon tube (Kasvi): 100 µL was used for tenfold serial dilutions and plating on SDA (CFU/mL), and 100 µL was used for biofilm dry weight assessment. The remainder volume (7.3 mL) was centrifuged (5500× *g*; 10 min; 4 °C), and washed three times with 2 mL of sterile milliQ water. The supernatants were pooled (13.3 mL) and the pellet was resuspended in 2 mL of milliQ water. Aliquots of the supernatant were used to quantify the proteins (1.1 mL), water-soluble polysaccharides (WSPs, 3 mL), and eDNA (650 µL). The pellet suspension was used to quantify the insoluble dry weight (500 µL), proteins (550 µL), alkali-soluble polysaccharides (ASPs, 750 µL), and lipids (200 µL).

#### 2.7.1. Biofilm Dry Weight

Aliquots (100 µL) of the biofilm scraped were transferred to microtubes that had been previously weighed. Samples were centrifuged (12,000 rpm; 15 min; 4 °C), the supernatant was discarded, and 300 µL of 99% ethanol was added. Then, samples were vortexed and kept at −80 °C for 18 h. Afterwards, the samples were centrifuged (12,000 rpm; 10 min; 4 °C), the supernatants were discarded, and the pellets were washed three times with 70% ethanol and air dried at room temperature. The microtubes were weighed again, and the difference between the weights was the biofilm dry weight [46,47]. To weight the samples, a precision weighing balance, with five decimal places (AUW 220D, Shimadzu, Tokyo, Japan), was used.

#### 2.7.2. Insoluble Dry Weight

A 500 µL aliquot of the pellet suspension was transferred to a previously weighed piece of aluminum foil. Each sample was dried at 100 °C, and then weighed again. The difference between the weights was the insoluble dry weight.

#### 2.7.3. Quantification of Soluble Proteins

A standard curve was prepared using bovine serum albumin (BSA; Sigma-Aldrich) at 0.0, 0.5, 1.0, 2.0, 4.0, 6.0, and 8.0 µg/mL in order to correlate protein concentrations with absorbance values (Appendix A). Aliquots of 500 µL from the biofilm supernatants and from each standard curve sample were individually mixed with 500 µL of Bradford reagent (Sigma-Aldrich) [48] in duplicate. After 20 min mixing at 70 rpm, absorbance was measured at 595 nm.

#### 2.7.4. Quantification of Insoluble Proteins

A 550 µL aliquot of the biofilm pellet suspension was heated at 100 °C for 90 min with mixing at 1,000 rpm, in order to extract its protein content. Then, 5 µL of the sample was added to 250 µL of Bradford reagent (Sigma-Aldrich) [48] in duplicate. Standard curves of BSA at 0.0, 0.5, 1.0, 2.0, 4.0, 6.0, and 8.0 µg/mL and at 0.0, 0.1, 0.2, 0.4, 0.6, 0.8, 1.0, and 1.2 mg/mL were also prepared. After 20 min mixing at 70 rpm, the absorbances of the test and standard curve samples were measured at 595 nm.

#### 2.7.5. Quantification of WSPs

A 3 mL aliquot of the biofilm supernatant was mixed with 7.5 mL of 99% ethanol. Samples were vortexed and kept at −80 °C for 18 h to precipitate the polysaccharides. Then, samples were vortexed, centrifuged (9500× *g*; 20 min; 4 °C), washed three times with 70% ethanol, and air dried at room temperature. The pellets were individually resuspended in 1 mL of water, and the total carbohydrate was quantified by the phenol-sulfuric acid method [49]. A standard curve using glucose at 0.00, 3.75, 6.25, 12.50, 25.00, 50.00, 75.00, 100.00, and 125.00 µg/mL was created (Appendix A). Aliquots of 200 µL of samples in triplicate and of each standard curve sample in duplicate were individually mixed with 200 µL of 5% phenol. Then, 1 mL of sulfuric acid was added to each sample. After 20 min, absorbance was measured at 490 nm.

#### 2.7.6. Quantification of ASPs

A 750 µL aliquot of the biofilm pellet suspension was transferred to a previously weighed microtube. Samples were then centrifuged (13,000× *g*; 10 min; 4 °C), the supernatants were discarded, and the pellets were dried in a sample concentrator (RVC 2-18C D, Martin Christ Gefriertrocknungsanlagen GmbH, Osterode, Germany) for 4.5 h and weighed. For ASP extraction, 300 µL of 1 N NaOH was added for each 1 mg of the sample’s dry weight. After 2 h of incubation, samples were centrifuged (13,000× *g*; 10 min), and the supernatants were individually collected. Another volume of 300 µL of 1 N NaOH/(mg of dry weight) was added to the pellets, vortexed, and incubated for 2 h. After centrifugation, the supernatant was collected and pooled with the previous supernatant. The same steps described were repeated with the pellet for a third time, but instead of being incubated for 2 h, samples were vortexed and centrifuged. The last supernatant was pooled with the previous supernatants and the pellet was discarded. Thereafter, three volumes of 99% ethanol were added to each sample, which was incubated at −20 °C for 18 h for ASP precipitation. After this time, samples were centrifuged (13,000× *g*; 20 min; 4 °C), the supernatants were discarded, and the pellets were washed three times with 70% ethanol as described for WSPs. The air-dried pellets were resuspended in the same volume of 1 N NaOH and used for quantification of total carbohydrate by the phenol-sulfuric acid method [49], as described for WSPs. A standard curve with 1 N NaOH and glucose at final concentrations of 0.00, 3.75, 6.25, 12.50, 25.00, 50.00, 75.00, 100.00, and 125.00 µg/mL was prepared (Appendix A).

#### 2.7.7. Quantification of eDNA

To extract eDNA from the samples, 650 µL of phenol:chloroform:isoamilic acid (25:24:1) was added to an aliquot of 650 µL of each biofilm supernatant [50]. Then, the samples were centrifuged (10,000 rpm; 5 min; 4 °C), and the supernatants were treated with chloroform:isoamilic acid (24:1). After centrifugation, the supernatants were transferred to another tube, and three volumes of 99% ethanol and 1/10 volume of 3 M sodium acetate were added. Samples were incubated at −20 °C for 18 h for DNA precipitation. The samples were then centrifuged (13,000× *g*; 20 min; 4 °C), and the pellets were washed three times with 70% ethanol, air-dried, and resuspended in Tris-EDTA (TE) buffer. The DNA concentration was measured in a nanospectrophotometer at 260 nm [50].

#### 2.7.8. Quantification of Lipids

A 200 µL aliquot of the biofilm pellet suspension was centrifuged (13,000× *g*; 10 min; 4 °C), the supernatant was removed, and the pellet was resuspended in 100 µL of 99% ethanol. Then, samples were vortexed, and the lipids were quantified by sulfo-phospho-vanillin reaction [51]. A standard curve using olive oil diluted in 99% ethanol at 0, 0.375, 0.750, 1.125, 2.250, 4.500, 7.500, and 9.750 mg/mL was prepared (Appendix A). Aliquots of 20 µL of each standard curve sample in duplicate were evaporated by incubation at 70 °C for 15 min. Then, 200 µL of sulfuric acid was added to each experimental sample and to each standard curve sample and vortexed. The test tubes were placed in boiling water for 10 min, and cooled in cold water for 5 min. Afterwards, 10 mL of the phospho-vanillin reagent was added to each tube, which was vortexed and incubated in boiling water for 15 min. The samples were cooled, and after 30 min absorbance was measured at 540 nm. 

### 2.8. Interaction of NAC with Antifungal Agents

The checkerboard microdilution method was used to evaluate the interaction of NAC with the antifungal agents nystatin (polyene), fluconazole (azole), and caspofungin (echinocandin) against planktonic cultures. Antifungal agents were purchased from Sigma-Aldrich and diluted (stock solutions) in dimethylsulfoxide (DMSO; for fluconazole and nystatin) or RPMId (caspofungin). Each drug solution was diluted 40× in the culture medium (RPMId) (work solution), so the final DMSO concentration was 2.5%. Twofold serial dilutions of NAC (50 µL) (in RPMId) and antifungal agents (50 µL) were distributed along the rows and columns, respectively, of a 96-well, U-bottom microtiter plate (Kasvi). The final concentrations of the drugs for both strains are shown in Table 1. Aliquots of 100 µL of CaS or CaR were added at the final concentration of 0.5–2.5×10^3^ CFU/mL. The control consisted of fungal inoculum with no drug (only the vehicle solution). After 24 h of incubation at 37 °C, the OD_540_ was measured; the control and samples whose OD_540_ values were lower than the control were diluted and plated on SDA for colony counting. The fractional inhibitory concentration index (FICI) was determined by the sum of the FICI of each agent (FICI = FICI_NAC_ + FICI_antifungal agent_). The FICI of each agent was calculated by dividing the MIC of the agent in combination by the MIC of the agent alone (FICI_A_ = MIC_A in the presence of B_/ MIC_A alone_). The interpretation of the FICI value was as follows: FICI < 0.5, synergism; 0.5 ≤ FICI ≤ 4.0, no interaction; and FICI > 4.0, antagonism [52]. Due to shortcomings of the FICI method [53], Bliss independence analysis [53,54,55] was performed, which is based on the idea that each drug acts independently following the probability of independent events. This model is described by the equation: E_IND_ = E_A_ + E_B_ − E_A_ × E_B_ for a combination of drug A at concentration *a* with drug B at concentration *b*. E_A_ is the percentage of growth inhibition observed for drug A alone at concentration *a*, E_B_ is the percentage of growth inhibition observed for drug B alone at concentration *b*, and E_IND_ is the expected percentage of growth inhibition of a non-interactive association of drug A at *a* with drug B at *b*. The difference (ΔE = E_OBS_ − E_IND_) between the observed percentage of growth inhibition (E_OBS_) and the expected one (E_IND_) defined the interaction of drugs for each concentration as follows: Bliss synergism, when ΔE and its 95% confidence interval (CI) were > 0; Bliss antagonism, when ΔE and its 95% CI were < 0; and Bliss independence, when 95% CI of ΔE overlapped 0 [53,54,55]. The experiments were performed in three independent occasions, FICI and Bliss independence analyses were performed for each combination of drugs for each occasion with the OD_540_ values, and mean values of ΔE were used to produce a three-dimensional, interaction surface plot, where peaks above the 0 correspond to synergism and valleys below the 0 correspond to antagonism, while the 0 plane itself indicates no statistically significant interaction. 

#### Interaction of NAC with Caspofungin against Biofilms

Biofilms from each strain were cultured for 48 h, as described before (Section 2.6), then the checkerboard assay was performed as described above. Only caspofungin was evaluated, since it was the only antifungal agent that showed synergism with NAC against planktonic cultures (see Results section). For both strains, the final concentrations of NAC (100 µL) were 50, 25, 12.5, and 0 mg/mL, and of caspofungin (100 µL) were 2, 1, 0.5, 0.25, 0.13, 0.06, 0.03, 0.02, 0.01, and 0 µg/mL. Controls received only the drug vehicle solution. After incubation at 37 °C for 24 h, samples were washed twice with PBS and the metabolic activity was evaluated by the 2,3-bis (2-methoxy-4-nitro-5-sulfophenyl)-5-[(phenylamino) carbonyl]-2H-tetrazolium hydroxide (XTT; Sigma-Aldrich) colorimetric assay. The XTT was prepared using ultrapure water at a concentration of 1 mg/mL and kept at −70 °C until needed. A menadione solution (Sigma-Aldrich) was prepared in acetone at 0.007 g/mL just prior to each experiment. The XTT solution prepared in all experiments consisted of PBS with 200 mM of glucose, XTT, and the menadione solution in the following proportions: 158 µL, 40 µL, and 2 µL, respectively. An aliquot of 200 µL of the XTT solution was added into each well with the biofilms, and incubated at 37 °C for 3 h. After this time, aliquots of 100 µL of the XTT solution degradation product (supernatant) were transferred to the wells of another 96-well, flat-bottom plate and absorbance was measured at 492 nm. Bliss independence analysis was performed as described above and illustrated using interaction surface plots.

### 2.9. Confocal Laser Scanning Microscopy (CLSM)

Biofilm viability and matrix composition were also analyzed with a confocal laser microscope (LSM 800 with Airyscan; Carl Zeiss, Germany). Biofilms were grown in duplicate in 24-well, flat-bottom microtiter plates (Kasvi), as described above for pre-formed biofilms, treated with or without (control) NAC at 100 mg/mL for 24 h, washed twice with PBS, stained according to the manufacturers’ instructions for each fluorochrome, and then samples were washed once with PBS before being observed by confocal laser scanning microscopy (CLSM). 

For viability analysis, samples were stained with the Live/Dead BacLight Bacterial Viability kit (Molecular Probes, Invitrogen Corp., Carlsbad, CA, USA). Syto9 and propidium iodide (excitation/emission at 488/488–550 nm and 488/656–700 nm, respectively) were diluted 1:1000 and incubated with the biofilm for 45 min in the dark. For polysaccharides (α-D-mannosyl and α-D-glucosyl) [56], concanavalin A-Alexa Fluor 594 (Molecular Probes) (561/600–700 nm) at 75 µg/mL was incubated with the biofilms for 30 min. For β-glucans [57], Calcofluor White (Sigma-Aldrich) (405/450 nm) at 0.5 g/L in 10% KOH was incubated with the biofilms for 10 min. For proteins [58], biofilms were incubated with FilmTracer Sypro Ruby (Molecular Probes) (488/576–700 nm) in the dark for 30 min. For lipids [59], Nile Red (Sigma-Aldrich) at 50 µg/mL prepared with 10 mL of 25% DMSO/H_2_O was incubated with the biofilm for 10 min (488/563 nm for neutral lipids and 561/613–700 nm for polar lipids). In order to distinguish the red fluorescence of Nile Red from the polar lipids, the green color was chosen for neutral lipids as an alternative to orange color. 

For each image, biofilm thickness was determined with *z*-stack readings at 1 µm. Quantification of each matrix component was estimated by the fluorescence intensity generated by each fluorochrome. For each biofilm sample (*n* = 2), 5 random fields were measured, totaling 10 fields for both duplicates.

### 2.10. Statistical Analysis

The MIC/MFC and biofilm assays were performed in quadruplicate on three different occasions (*n* = 12 for each group). The values of CFU/mL were log_10_ transformed and outliers were removed. For the matrix composition of biofilms, the sample size (α = 5%, power = 80%) was calculated using MedCalc software (version 19.1; MedCalc Software, Mariakerke, Belgium), resulting in *n* = 6. For each assay, data obtained were rated according to their original volume (biofilm supernatant or biofilm pellet suspension). Biofilm dry weight was used to normalize the data from WSPs, soluble proteins, and eDNA, while insoluble dry weight was used to normalize the data from ASPs, lipids, and insoluble proteins. Normal distribution and homoscedasticity of data were evaluated by Shapiro–Wilk and Levene tests, respectively. Data were analyzed by two-way ANOVA (with strain and treatment as independent variables). For homoscedastic data, a post-hoc Tukey test was used. When data were heteroscedastic, they were evaluated by a post-hoc Games–Howell test. The level of significance was 5% and SPSS software (version 20.0; SPSS Inc., Chicago, IL, USA) was used. Data from the checkerboard assays and CLSM were analyzed as previously described.

## 3. Results

### 3.1. Antifungal Evaluation of NAC

The MIC of NAC was 25 mg/mL after 24 h and 48 h for both strains. After 24 h, a significant interaction (*p* < 0.001) was found between the strain and the treatment (Figure 1A), while after 48 h, no interaction (*p* = 0.915) was observed, but a significant effect (*p* ≤ 0.04) was verified for both the treatment (Figure 1B) and the strain (mean ± SD of 0.98 ± 0.79 au, for CaS and 0.79 ± 0.74 au for CaR). No MFC was found, since no concentration of NAC inhibited the colony growth of the strains. However, significant reduction of fungal viability was observed. After 24 h, there was a significant interaction (*p* = 0.008) between the strain and the treatment. NAC at 50 and 25 mg/mL reduced (*p* < 0.001) the viability of CaS by 4.06 and 2.25 log_10_, respectively, and of CaR by 2.98 and 1.81 log_10_, respectively. Conversely, NAC at 12.5 mg/mL increased (*p* ≤ 0.041) the fungal viability by 0.34 and 0.57 log_10_ for CaS and CaR, respectively (Figure 1C). After 48 h of incubation with NAC, two-way ANOVA demonstrated a significant interaction (*p* = 0.001) between the strain and the treatment. Only the concentration of 50 mg/mL reduced (*p* < 0.001) the viability of CaS by 2.26 log_10_ (Figure 1D), while NAC at 50 and 25 mg/mL reduced (*p* < 0.001) the colony growth of CaR by 3.26 and 0.46 log_10_, respectively. 

### 3.2. Time-Kill Curves

For both strains, OD_540_ values increased after 6 h (Figure 2) for control samples and those treated with NAC at concentrations lower than the MIC (subMIC). NAC at 25 and 50 mg/mL inhibited fungal growth during the 24 h incubation, but after 22 h the samples treated with 25 mg/mL NAC showed a slight increase in the OD_540_ values. Compared with the controls (no NAC), samples with NAC at subMIC demonstrated higher OD_540_ values, suggesting that these concentrations increased fungal growth.

### 3.3. Evaluation of the Effect of NAC on Biofilm Formation

The results of viability assessments demonstrated a significant interaction (*p* < 0.001) between the strain and the treatment. Compared with the controls, NAC at 100 and 50 mg/mL inhibited (*p* ≤ 0.005) biofilm formation by 1.63 and 1.42 log_10_ for CaS, respectively, and by 2.77 and 1.52 log_10_ for CaR, respectively (Figure 3A). However, NAC at 12.5 mg/mL increased the biofilm formation of CaS by 0.73 log_10_ (*p* = 0.01).

When total biomass was evaluated, there was significant interaction (*p* < 0.001) between the strain and the treatment. NAC at 100, 50, and 25 mg/mL reduced (*p* < 0.001) the biofilm formation of CaS by 65.13, 88.70, 77.33%, respectively, compared with the control. For CaR, only NAC at 100 mg/mL reduced the total biomass of biofilms, by 63.62% (*p* = 0.003) (Figure 3B).

### 3.4. Effect of Incubation Time with NAC on Biofilm Formation

At the late stage (24 h) of biofilm formation in the presence of NAC, two-way ANOVA showed a significant interaction (*p* = 0.001) between the strain and the treatment. Only NAC at 100 mg/mL inhibited (*p* < 0.001) the CaS viability, by 1.98 log_10_, during biofilm formation (Figure 3C). For CaR, NAC at 100 and 50 mg/mL reduced (*p* < 0.001) the fungal viability by 1.71 and 1.09 log_10_, respectively (Figure 3C). For the total biomass, a significant interaction (*p* = 0.015) was found between the strain and the treatment. NAC at 100 and 50 mg/mL reduced the CaS total biomass by 85.62 and 58.79% (*p* ≤ 0.002), respectively, and for CaR only NAC at 100 mg/mL reduced the total biomass, by 52.63% (*p* = 0.041) (Figure 3D).

When biofilm formation was evaluated during the middle stage (12 h), a significant interaction was observed between the strain and the treatment for viability (*p* = 0.004) and total biomass (*p* = 0.014). NAC at 100 and 50 mg/mL reduced the CaS viability by 0.87 and 0.62 log_10_ (*p* ≤ 0.001), respectively, and the CaR viability by 0.94 and 1.42 log_10_ (*p* ≤ 0.008), respectively (Figure 3E). These concentrations also reduced the total biomass of CaS biofilm by 67.13 and 69.42% (*p* < 0.001), respectively, and of CaR biofilm by 60.42 and 62.11% (*p* ≤ 0.003), respectively (Figure 3F).

At the early stage (6 h) of biofilm formation, there was also a significant interaction between the strain and the treatment for viability (*p* = 0.022) and total biomass (*p* < 0.001). NAC at 100 and 50 mg/mL inhibited the CaS viability by 1.10 and 1.09 log_10_ (*p* < 0.001), respectively (Figure 3G), and the CaR viability by 1.34 and 1.56 log_10_ (*p* < 0.001), respectively. Moreover, NAC at 25 mg/mL also reduced (*p* < 0.001) the CaR viability by 0.70 log_10_ compared with the control (Figure 3G). For the total biomass after 6 h, NAC at 100, 50, and 25 mg/mL inhibited the biofilm formation by 77.37, 81.47, and 39.98%, respectively, for CaS (*p* ≤ 0.006), and by 86.28, 87.35, and 79.77%, respectively, for CaR (*p* < 0.001) (Figure 3H).

### 3.5. Evaluation of the Effect of NAC on Pre-Formed Biofilms

When viability was assessed, a significant interaction (*p* < 0.001) between the strain and the treatment was verified. Only NAC at 100 mg/mL decreased (*p* < 0.001) the CaS biofilm, by 1.14 log_10_, compared with the control (Figure 4A). For CaR, NAC at 100, 50, and 25 mg/mL reduced (*p* ≤ 0.025) the CFU/mL values by 2.30, 1.26, and 0.64 log_10_, respectively (Figure 4A). Conversely, NAC at 12.5 mg/mL increased (*p* < 0.001) the viability of CaS biofilm by 0.26 log_10_. The findings of the biofilm total biomass evaluation also demonstrated a significant interaction (*p* < 0.001) between the strain and the treatment. Only NAC at 100 mg/mL decreased (*p* < 0.001) the CaS total biomass, by 73.69%. For CaR, NAC at 100, 50, and 25 mg/mL decreased (*p* ≤ 0.026) its total biomass by 69.28, 63.54, and 47.53%, respectively (Figure 4B).

### 3.6. Effect of NAC on the Composition of the Biofilm Matrix

When pre-formed biofilms of CaS and CaR were treated with NAC, two-way ANOVA demonstrated no significant interaction (*p* = 0.837) between the strain and the treatment for viability, and a significant effect was observed only for the treatment (*p* < 0.001). NAC at 100 mg/mL decreased (*p* < 0.001) the pre-formed biofilms by 0.60 log_10_ (Figure 5A). For biofilm formation in the presence of NAC, a significant interaction between the strain and the treatment was observed (*p* < 0.001) for viability. While NAC at 50 mg/mL reduced (*p* ≤ 0.01) the fungal viability by 0.78 and 1.36 log_10_ for CaS and CaR, respectively (Figure 5B), NAC at 12.5 mg/mL increased (*p* < 0.001) the CaS viability by 0.33 log_10_.

#### 3.6.1. Biofilm Dry Weight

For pre-formed biofilms, there was no significant interaction (*p* = 0.803) between the strain and the treatment, and a significant effect was found only for the treatment (*p* = 0.004). Biofilms treated with NAC at 100 mg/mL showed a significant (*p* = 0.003) reduction of dry weight by 6.38 mg (Figure 5C). For the biofilm formation analysis, no significant interaction (*p* = 0.23) was found between the strain and the treatment, but a significant effect (*p* < 0.001) was observed for both the strain and treatment. The biofilm dry weight of CaS (mean ± SD of 25.42 ± 19.42 mg) was higher (*p* < 0.001) than that of CaR (19.42 ± 9.79 mg). NAC at 50 mg/mL reduced (*p* < 0.001) the biofilm dry weight by 12.13 mg, while NAC at 12.5 mg/mL increased (*p* < 0.001) the biofilm dry weight by 11.13 mg (Figure 5D).

#### 3.6.2. Insoluble Dry Weight

No interaction (*p* = 0.428) between the strain and the treatment was observed for pre-formed biofilms, and only treatment was a significant factor (*p* = 0.034). NAC at 100 mg/mL decreased the insoluble dry weight (*p* = 0.037) by 3.03 mg (Figure 5E). For the biofilm formation analysis, a significant interaction (*p* < 0.001) was verified between the strain and the treatment. NAC at 50 mg/mL decreased the insoluble dry weight only for CaR (*p* = 0.004), by 3.33 mg (Figure 5F), while 12.5 mg/mL NAC increased the insoluble dry weight during biofilm formation of CaS (*p* = 0.001) by 2.29 mg (Figure 5F).

#### 3.6.3. Quantification of Soluble Proteins

Two-way ANOVA showed a significant interaction (*p* = 0.045) between the strain and the treatment for pre-formed biofilms. NAC at 100 mg/mL decreased (*p* = 0.036) soluble proteins of CaS by 95.62 µg/mL, but not of CaR (*p* = 0.217) (Figure 6A). For the biofilm formation analysis, a significant interaction (*p* = 0.015) was found between the strain and the treatment. However, no significant difference (*p* ≥ 0.084) between control and NAC at 50 and 12.5 mg/mL was observed for either strain (Figure 6B).

#### 3.6.4. Quantification of Insoluble Proteins

In this assay, some samples treated with 100 and 50 mg/mL NAC for pre-formed biofilms and biofilm formation, respectively, showed OD_595_ values below the lowest point on the standard curve. However, the samples’ concentrations were calculated based on the standard curve, which was used to calculate the original sample volume. However, the data showed a wide variation, and a statistical inference could not be made (Appendix A). 

#### 3.6.5. Quantification of WSPs

For pre-formed biofilm, there was a significant interaction between the strain and the treatment (*p* = 0.003). NAC at 100 mg/mL decreased WSPs (*p* ≤ 0.017) by 2696.19 and 798.15 µg/mL for CaS and CaR, respectively (Figure 6C). For the biofilm formation analysis, no significant interaction (*p* = 0.056) was found between the strain and the treatment, but a significant effect was verified for the strain (*p* = 0.001) and for the treatment (*p* = 0.015). The level of WSPs for CaS biofilms (mean ± SD of 1468.16 ± 1471.80 µg/mL) was higher than that for CaR biofilms (325.25 ± 169.63 µg/mL). No significant difference (*p* ≥ 0.097) was observed between biofilms treated with NAC and the control (Figure 6D). 

#### 3.6.6. Quantification of ASPs

For pre-formed biofilm, no significant interaction was observed between the strain and the treatment (*p* = 0.858), and neither of the main factors showed a significant effect (*p* ≥ 0.106, mean ± SD of 1099.57 ± 103.15 and 1677.94 ± 461.99 µg/mL for CaS and CaR, respectively, treated with 100 mg/mL NAC; and 1401.34 ± 968.14 and 2117.98 ± 1571.12 µg/mL for controls CaS and CaR, respectively). However, for the biofilm formation analysis, a significant interaction was found between the strain and the treatment (*p* = 0.024). NAC at 50 mg/mL reduced (*p* = 0.035) ASPs by 1538.16 µg/mL only for CaR (Figure 6E). 

#### 3.6.7. Quantification of eDNA

Two-way ANOVA demonstrated no significant interaction between the strain and the treatment (*p* = 0.152) for pre-formed biofilm, and only the treatment was a significant factor (*p* = 0.021). NAC at 100 mg/mL reduced the eDNA (*p* = 0.028) in 1008.73 ηg/µL (Figure 6F). For the biofilm formation analysis, there was no significant interaction between the strain and the treatment (*p* = 0.269), and only the strain was a significant factor (*p* = 0.01). CaS showed higher (*p* = 0.017) values of eDNA (mean ± SD of 2915.18 ± 1718.37 ηg/µL) than CaR (1354.24 ± 529.91 ηg/µL).

#### 3.6.8. Quantification of Lipids

For pre-formed biofilms, no significant interaction between the strain and the treatment (*p* = 0.245) was verified, and neither of the main factors demonstrated a significant effect (*p* ≥ 0.661, mean ± SD of 33.62 ± 6.23 and 37.24 ± 8.12 mg/mL for CaS and CaR, respectively, treated with 100 mg/mL NAC; and 36.98 ± 14.51 and 29.92 ± 12.72 mg/mL for controls CaS and CaR, respectively). For biofilm formation analysis, no significant interaction was observed between the strain and the treatment (*p* = 0.051), but both main factors showed a significant effect (*p* ≤ 0.036). CaR showed higher (*p* = 0.036) values of lipids (mean ± SD of 36.27 ± 29.03 mg/mL) than CaS (23.33 ± 12.33 mg/mL). NAC at 50 mg/mL showed higher (*p* < 0.001) values of lipids than 12.5 mg/mL NAC and the control (Figure 6G). 

### 3.7. Interaction of NAC with Antifungal Agents

The MICs of nystatin, fluconazole, and caspofungin for CaS were 2, 0.5, and 0.25 µg/mL, respectively, and for CaR were 2, 125, and 0.5 µg/mL, respectively. The FICI method showed synergism of NAC only with caspofungin for CaS, with FICI values of 0.245 (3.125 mg/mL NAC with 0.015 µg/mL caspofungin) and 0.373 (3.125 mg/mL NAC with 0.031 µg/mL caspofungin), and reductions of viability from 1.15 to 1.28 log_10_, respectively (Appendix A). However, the Bliss independence analysis demonstrated antagonism of NAC with nystatin for CaS (Figure 7A, CI from −0.6803–−0.0003 to −1.6437–−1.0301 for NAC at all concentrations with 2 µg/mL nystatin) and CaR (Figure 7B, CI from −0.2059–−0.0398 to −1.2331–−0.5081 for NAC at 25 and 6.25 mg/mL, respectively, with 2 µg/mL nystatin). For CaS, synergism was observed for NAC at 1.56 mg/mL with fluconazole at 0.125 µg/mL (Figure 7C, CI 0.0084–0.1115). For CaR, synergism was observed with NAC at 0.78 mg/mL with fluconazole at 15.62, 7.81, 0.97, and 0.48 µg/mL (Figure 7D, CI from 0.0684–0.1808 to 0.0720–0.3834), but antagonism was also observed for NAC at 6.25 mg/mL with fluconazole at 1.95 µg/mL (CI −0.2277–−0.0340). Synergism with higher values of mean ΔE was observed for CaS when NAC at 6.25 and 3.125 mg/mL was associated with caspofungin from 0.062 to 0.015 µg/mL (Figure 7E, CI from 0.0141–0.1560 to 0.0892–2.2102). For CaR, synergism was also verified with NAC at 3.125 and 1.56 mg/mL associated with 0.25 µg/mL caspofungin (Figure 7F, CI from 0.0035–0.1328 to 0.6138–0.6496). Bliss independence analysis was not performed for CFU/mL data, because only samples with OD_540_ values lower than the control were plated (Appendix A).

#### Interaction of NAC with Caspofungin against Biofilms

All concentrations of caspofungin alone (with NAC at 0 mg/mL) resulted in the highest values of metabolic activity for biofilms of both strains. Therefore, FICI was not calculated. NAC alone at 50 and 25 mg/mL (with caspofungin at 0 µg/mL) reduced the metabolic activity of CaS biofilms by 37.37% and 24.44%, respectively, and of CaR biofilms by 59.29% and 57.69%, respectively. NAC at 12.5 mg/mL with caspofungin at 0.03 µg/mL reduced the metabolic activity of CaS and CaR biofilms by 85.05% and 49.45%, respectively. The Bliss independence analysis showed antagonism for CaS (Figure 8A, CI from −0.1421–−0.1391 to −1.1360–−0.8701), but synergism for CaR (Figure 8B, CI from 0.0257–0.0408 to 0.1698–0.4252).

### 3.8. CLSM

CLSM images (Figure 9 and Figure 10) showed that biofilms treated with NAC were thinner than those of the controls for viability, polysaccharide, lipid, and protein experiments for both strains, with the exception of CaS viability (Table 2). When fluorescence intensity was quantified (Table 2), control biofilms showed higher values compared with biofilms treated with NAC for living cells, polysaccharides (α-D-mannosyl and α-D-glucosyl), and polar lipids for both strains; it also demonstrated a higher intensity of dead cells when biofilms were treated with NAC. Conversely, for polysaccharides (β-glucans), proteins, and neutral lipids, the fluorescence intensities of control biofilms were lower than those treated with NAC for CaS, while CaR showed lower values for the biofilms treated with NAC.

## 4. Discussion

Our investigation demonstrated that NAC showed the same MIC (25 mg/mL) for CaS and CaR, an important finding in the current battle against antimicrobial resistance. Although we did not evaluate the target of NAC in the fungal cell to explain its fungicidal action, one of the limitations of our study, we suggest that NAC may act on the polysaccharides of the fungal cell wall, which is composed of mannan, glucan and chitin [4]. Such hypothesis is supported by our findings from the matrix components assays (reduction of soluble polysaccharides) and may also explain the same susceptibility of both fluconazole-resistant and -susceptible strains to NAC. Although other studies have shown an antifungal effect of NAC against *C. albicans*, only clinical isolates were used [37,40,41] and only one study reported that the strains were susceptible to ketoconazole [41]. Therefore, the present investigation may be the first to report the antifungal effect of NAC on a resistant strain of *C. albicans*. Moreover, this investigation evaluated reference strains since, as was observed by Alnuaimi et al. (2013) [60], reference strains have higher biofilm growth activity than clinical isolates, which is important for an in vitro evaluation before an in vivo evaluation. The studies mentioned above showed NAC MIC values of 40, 20, and 16.5 mg/mL against clinical isolates of *C. albicans* using the macrodilution and agar dilution methods [40,41]. The differences between the results of our investigation and those of the mentioned studies [35,36] may be ascribed to the different stock concentrations of NAC and the strains used. In all these investigations, the MIC values were close to the NAC stock concentrations used, which demonstrates that high concentrations are needed for a fungistatic effect. However, a fungicidal effect (MFC) was not observed in our investigation, since none of the concentrations of NAC inhibited colony growth. Another study showed that NAC at 160 mg/mL inhibited 99.9% of fungal growth [41]. However, inhibition of microbial growth should be evaluated by logarithm, since only 1 log of reduction is the same as a 99% decrease. According to the American Society for Microbiology, an agent is considered as antimicrobial when it reduces the microbial growth by at least 3 log [61]. In the present investigation, significant reductions from 1.81 to 4.06 log were observed. Therefore, although an MFC was not found, NAC was effective in reducing the fungal viability of both strains.

The time-kill curves showed that NAC at the MIC or higher concentrations reduced the fungal growth after 24 h; this was more evident after 6–10 h. Another study also verified that NAC at 80 mg/mL decreased *C. albicans* viability after 24 h, while lower concentrations resulted in an increase of fungal viability after 2–4 h [40]. The present investigation also found an increase of fungal growth when NAC was used at subMIC levels. NAC is a precursor of glutathione, an essential metabolite for *C. albicans*; at low concentrations NAC may provide glutathione to *C. albicans*, which is metabolized via γ-glutamylcysteine synthetase—GCS1 [62]. Thus, the results suggest that NAC at subMIC levels may stimulate fungal growth, but this in vitro finding cannot be extrapolated to the clinical scenario. 

As a preventive strategy to control biofilm development, NAC was added during biofilm formation. NAC at concentrations higher than the MIC inhibited biofilm formation for both strains tested, reducing both the viability and the biomass of the biofilms. This result was not surprising, since the MIC assay demonstrated that these concentrations were fungistatic. Another study showed a reduction of biofilm formation in clinical *C. albicans* isolates using NAC at subMIC [41]. However, in this study, NAC at subMIC (12.5 mg/mL) showed a similar result to the control biofilm (without NAC) for CaR, while NAC at 12.5 mg/mL increased biofilm viability for CaS. These findings confirm that NAC at low concentrations may have an opposite effect to an antimicrobial agent. When the incubation time was evaluated, NAC at the MIC (25 mg/mL) reduced biofilm formation of CaR at the early stage (6 h), while NAC at concentrations higher than the MIC decreased the biofilm formation at all stages (6, 12, and 24 h) of both strains, except for the viability of CaS biofilms incubated with 50 mg/mL NAC for 24 h. Although NAC at the MIC was not evaluated for the intermediate–late stages (12 and 24 h), the findings observed in the incubation time assay corroborated those found for 48 h biofilm formation, in which only NAC at 100 and 50 mg/mL reduced biofilm formation. 

As NAC is a mucolytic agent, it was tested against pre-formed biofilms as a therapeutic strategy for disrupting biofilms. Only NAC at 100 mg/mL reduced the viability and biomass of CaS biofilm, while other concentrations (50, and 25 mg/mL, as well as 100 mg/mL) could also decrease the viability and biomass of CaR biofilm. Therefore, NAC was more effective for pre-formed biofilms of CaR. Other studies also have observed a reduction of the viability and biomass of pre-formed biofilms after treatment with NAC [37,41]. However, these studies did not evaluate resistant strains. The present investigation also demonstrated a significant increase in the biofilm viability when NAC was used at 12.5 mg/mL. However, clinical isolates may have a different susceptibility response. Therefore, since we did not evaluate clinical isolates, it may be a limitation of our study.

A novel aspect of this investigation is the evaluation of the effect of NAC on the matrix composition of fungal biofilms. Before the quantification of the matrix components, the viability of biofilms treated (pre-formed biofilms) and grown with NAC (biofilm formation in the presence of NAC) was evaluated. The reductions observed for pre-formed biofilms treated with NAC at 100 mg/mL was lower than those observed in the previous assay using 96-well microtiter plate (see Section 3.5). Probably the larger area of 24-well microtiter plate and volume of fungal inoculum used in the matrix assays could have improved the biofilm grown and reduced its susceptibility to NAC. On the other hand, for biofilm formation, the reductions observed in the presence of NAC at 50 mg/mL and the increase of CaS with NAC at 12.5 mg/mL agreed with those found the previous assay (see Section 3.3). The results also showed that NAC reduced the soluble components of the biofilm matrix (soluble proteins for CaS, and WSPs and eDNA for both strains) for pre-formed biofilms, but not for biofilm formation. Other investigations have evaluated the effect of NAC only on the extracellular polysaccharides of the biofilm supernatant (WSPs) of bacterial strains [31,38]. In those studies, NAC inhibited WSP production at lower concentrations (0.5 and 1 mg/mL) than those needed for viability inhibition [31,38]. In this study, only concentrations of NAC higher than the MIC were used to evaluate the matrix components, since NAC at subMIC did not reduce biofilm formation and pre-formed biofilms in the previous assays. Thus, NAC at subMIC would be effective in reducing the WSPs only for bacterial biofilms. Although NAC reduced the viability, the biofilm dry weight, and the insoluble dry weight of both pre-formed biofilms and of biofilms during their formation in the present investigation, it was able to reduce the soluble components of the matrix only for pre-formed biofilm. NAC had no effect on the soluble components during biofilm formation. This result suggests that NAC is able to disrupt pre-formed biofilms by acting on their soluble components, but does not interfere in the synthesis of the matrix components during biofilm formation. 

NAC was ineffective in reducing most water-insoluble components (ASPs and lipids), with the exception of the ASPs of CaR during biofilm formation. A previous study evaluated the effect of an antifungal agent (fluconazole) on the biofilm matrix of the same resistant strain used in the present investigation. Using the same methodology to quantify the matrix compounds, the authors observed reduction of WSPs and biomass (total dry weight) for all strains evaluated, and reduction of ASPs only for CaS [63]. Another investigation demonstrated higher values of ASPs than WSPs from biofilms of CaS treated with fluconazole and nystatin [64]. All these findings demonstrate that the insoluble components of the biofilm matrix may be less susceptible to antibiofilm approaches.

The CLSM analysis corroborated, in part, the biochemical quantification assays of the matrix components. Biofilms treated with NAC showed a lower thickness than the control biofilms, except for those of CaS stained with Live/Dead (for viability analysis). This unexpected result may be ascribed to the dense cells observed for biofilms of CaS (Figure 9) compared with those of CaR (Figure 10), which may hinder the penetration of NAC into the CaS biofilm. Moreover, the fluorescence intensity results for biofilms treated with NAC also showed lower values than for the control biofilms, except for polysaccharides (β-glucans), proteins, and neutral lipids of CaS biofilms. These exceptions may be justified by the insolubility of these components in water, as observed in the matrix-component quantification assays. However, beyond the lower thickness of samples treated with NAC, the images clearly show more “black spaces” in the samples treated with NAC compared with controls. In addition, hyphae were observed only in biofilms stained with Sypro Ruby (proteins). Since the biofilms were grown with an RPMI-based medium, it was expected that we would observe a predominance of hyphae in the samples. It is likely that the supplementation of the medium with glucose and the staining procedure used for the other chromophores may have influenced the yeast morphology of *C. albicans*. Another study also used Calcofluor White and Sypro Ruby to evaluate, respectively, the architecture and the permeability of the biofilms of two types of *C. albicans*: a heterozygous strain of the mating type locus (MTL; a/α cells) and an MTL-homozygous strain (a/a cells). Although biofilms formed by both kind of cells showed similar architecture, a/α biofilms showed pathogenic traits of impermeability, impenetrability by human phagocytic leukocytes, and fluconazole-resistance, while a/a biofilms did not show these traits [65].

According to Nett et al. (2007) [66] and Zarnowski et al. (2014) [12], the WSP comprises the α-mannans, and the ASP comprises 1,6-β-glucan and 1,3-β-glucan. Mannan is indicated as the most abundant polysaccharide in the fungal biofilm matrix, comprising approximately 87% of the polysaccharide, followed by 1,6-β-glucan at 13%, and a tiny percentage of 1,3-β-glucan, which together give rise to the mannan-glucan complex (MGC) [12]. Another investigation suggested that the stability of each matrix polysaccharide (α-mannans, 1,6-β-glucan, and 1,3-β-glucan) is responsible for maintaining the MGC, which is crucial for biofilm resistance [67]. Moreover, the 1,3-β-glucan in the biofilm is also responsible for drug sequestration and fluconazole resistance [66,68]. Consequently, a reduction in any polysaccharide of the matrix may increase the susceptibility of the biofilm to antifungal agents. When the enzyme 1,3-β-glucanase was used against *C. albicans*, a reduction in the viability of the biofilm but not the planktonic culture was observed [66]. Moreover, other enzymes promoted detachment of *C. albicans* biofilms from a polystyrene surface, such as proteinase K, chitinase, DNAse I, β-*N*-acetylglucosaminidase, and lyticase, while lipase type II, phospholipase A2, and protease type XIV did not [13]. Lyticase, which also hydrolases 1,3-β-glucan, caused the greatest reduction of the biofilms [13].

When associated with antifungal agents, NAC showed greater synergism with caspofungin for planktonic cultures, which may be explained by the similar mechanism of action of the two drugs. Since caspofungin inhibits 1,3-β-D-glucan synthase, an enzyme necessary for the synthesis of an important fungal cell-wall polysaccharide [69], and NAC is reported to reduce the polysaccharides of biofilm matrix [31,38], as observed in the quantification of WSPs of pre-formed biofilms in the present investigation, the association of both drugs may have potentiated the antifungal effect. However, this association showed antagonism for CaS biofilms, which suggests that the biofilm alters the action of this combination of drugs, although higher decrease of metabolic activity of CaS was observed when the drugs were associated than that observed with NAC alone. Moreover, antagonism was also observed for NAC associated with nystatin, suggesting that the mechanism of action of one or both drugs may be hindered when the drugs are combined. Although the combination of NAC with fluconazole showed synergism at some concentrations, this result should be viewed with caution, since low mean values of ΔE (percentage growth inhibition) were observed (only 6% for CaS and 12% to 23% for CaR), which may be not enough to properly treat an infection caused by *C. albicans*. 

In another study [37], NAC had a synergistic effect with amphotericin B and fluconazole against biofilms of *C. albicans*. However, the authors did not evaluate planktonic cultures and the synergism was evaluated by the median effect, which employs equations to calculate the combination indices. Furthermore, NAC and fluconazole did not show synergism when the combination ratio was 1000:1, evaluated by an inhibitory effect of 90% [37]. In the present investigation, we did not evaluate the interaction of NAC with another polyene (nystatin) and fluconazole in biofilms, because the FICI showed no synergism of these drugs in the planktonic assays. Another study demonstrated that the association of NAC with ketoconazole promoted higher inhibition of biofilm adherence and pre-formed biofilms than the drugs alone [41]. Synergism was also observed when fluconazole and amphotericin B were associated with the enzyme 1,3-β-glucanase against *C. albicans* biofilm [66]. However, the mentioned studies did not use the Bliss independence method for synergism evaluation.

It is important to note that NAC is a drug commercially available for clinical use, especially for respiratory illnesses, and, therefore, it is safe. A report reviewing the multiple clinical applications of NAC pointed out that even doses up to 1200 mg are well tolerated and have unusual side effects [70]. This information gave us confidence regarding the use of NAC in our research, because the highest concentration tested was 100 mg/mL. For this reason, we did not evaluate the cytotoxic effect of NAC, which is another limitation of our study. However, as already mentioned, the clinical use of NAC is well established, and consequently it does not show side effects when properly used.

In conclusion, NAC showed a fungistatic effect against CaS and CaR. However, a subMIC level (12.5 mg/mL) resulted in the opposite effect (an increase of fungal growth). The antibiofilm effect of NAC was due to its fungistatic action, since only concentrations greater than the MIC reduced biofilm formation, pre-formed biofilms, biofilm thickness, and the matrix composition, especially the water-soluble components of pre-formed biofilms. NAC also showed synergism with caspofungin for planktonic cultures, but not for biofilms of CaS. However, clinical isolates were not used in this investigation, which may have different virulence factors and susceptibility to NAC. Therefore, our in vitro findings cannot be extrapolated to the clinical scenario. Future studies are needed to elucidate the fungistatic mechanism of action of NAC against *C. albicans*, and to evaluate the efficacy of NAC against other non-*albicans Candida* species and the infections caused by them. 

## Figures and Tables

**Figure 1 microorganisms-08-00980-f001:**
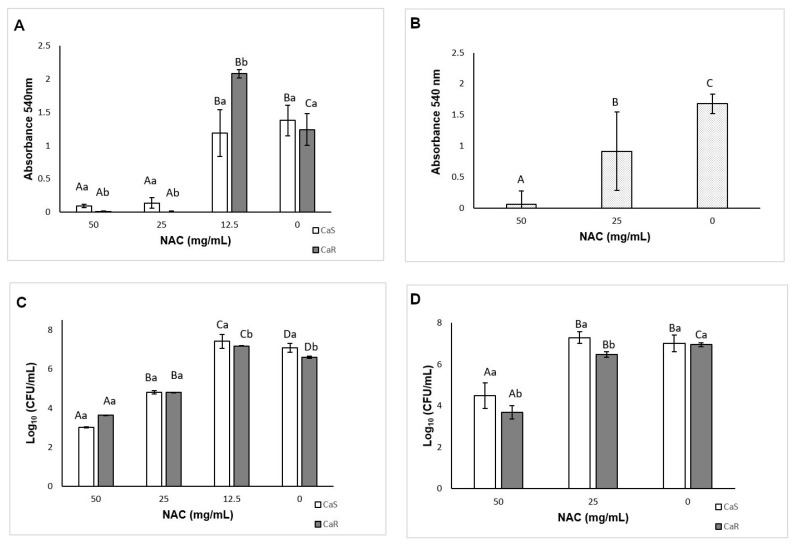
Mean optical density (OD) values obtained for (**A**) 24 h minimum inhibitory concentration (MIC) against *Candida albicans* susceptible (CaS) and *Candida albicans* resistant (CaR) and for (**B**) 48 h MIC against CaS and CaR; mean log_10_(CFU/mL) values for (**C**) 24 h CaS and CaR and (**D**) for 48 h CaS and CaR. Error bars: standard deviation (*n* = 12). Different capital letters (A, B, C) on the top of the bars denote a significant difference (*p* < 0.05) among the *N*-acetylcysteine (NAC) concentrations for the same strain, and different lowercase (a, b) letters denote a significant difference (*p* < 0.05) between the strains for the same NAC concentration. In the same way, similar capital letters on the top of the bars denote no significant difference (*p* ≥ 0.05) among the groups for the same strain, and similar lowercase letters denote no significant difference (*p* ≥ 0.05) between the strains for the same NAC concentration.

**Figure 2 microorganisms-08-00980-f002:**
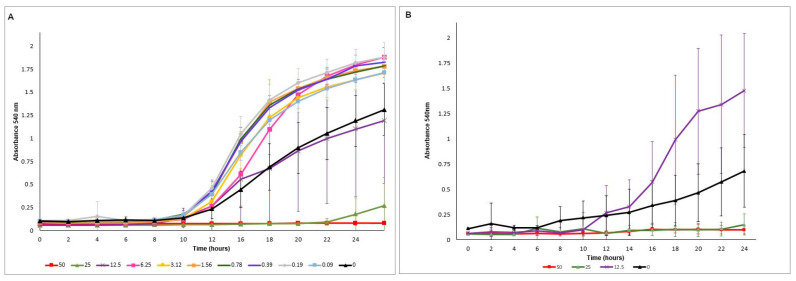
Time-kill curves, mean absorbance (OD_540_) values obtained for CaS (**A**) and CaR (**B**) grown with NAC during 24 h. Error bars: standard deviation (*n* = 12).

**Figure 3 microorganisms-08-00980-f003:**
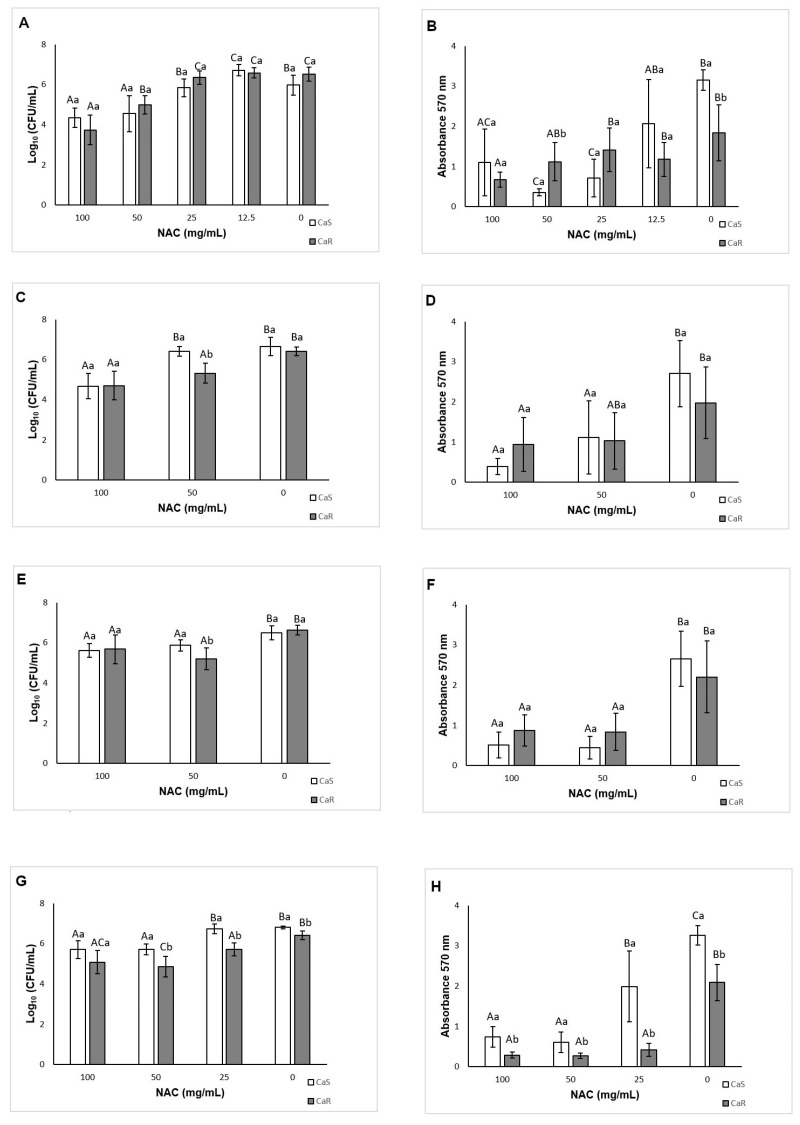
Mean log_10_(CFU/mL) and absorbance (total biomass) values for CaS and CaR obtained during biofilm formation in the presence of NAC at 48 h (**A**,**B**), 24 h (**C**,**D**), 12 h (**E**,**F**), and 6 h (**G**,**H**). Error bars: standard deviation (*n* = 12). Different capital letters (A, B, C) on the top of the bars denote a significant difference (*p* < 0.05) among the NAC concentrations for the same strain, and different lowercase (a, b) letters denote a significant difference (*p* < 0.05) between the strains for the same NAC concentration. In the same way, similar capital letters on the top of the bars denote no significant difference (*p* ≥ 0.05) among the groups for the same strain, and similar lowercase letters denote no significant difference (*p* ≥ 0.05) between the strains for the same NAC concentration.

**Figure 4 microorganisms-08-00980-f004:**
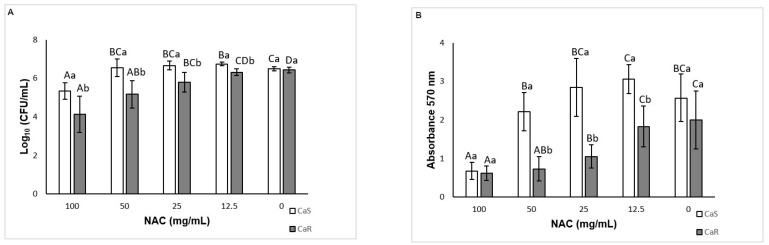
Mean log_10_(CFU/mL) (**A**) and absorbance (total biomass) values (**B**) for CaS and CaR obtained for pre-formed biofilm in the presence of NAC. Error bars: standard deviation (*n* = 12). Different capital letters (A, B, C) on the top of the bars denote a significant difference (*p* < 0.05) among the NAC concentrations for the same strain, and different lowercase (a, b) letters denote a significant difference (*p* < 0.05) between the strains for the same NAC concentration. In the same way, similar capital letters on the top of the bars denote no significant difference (*p* ≥ 0.05) among the groups for the same strain, and similar lowercase letters denote no significant difference (*p* ≥ 0.05) between the strains for the same NAC concentration.

**Figure 5 microorganisms-08-00980-f005:**
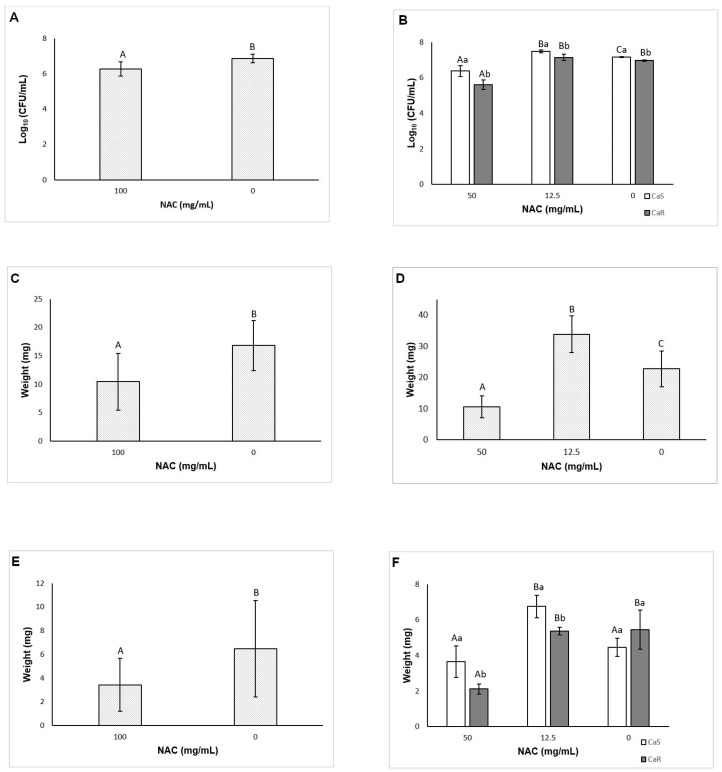
Mean log_10_(CFU/mL) values for pre-formed biofilm (**A**) and for biofilm formation (**B**). Mean values of biofilm dry weight of pre-formed biofilms (**C**) and for biofilm formation (**D**). Mean values of insoluble dry weight of pre-formed biofilm (**E**) and for biofilm formation (**F**). Error bars: standard deviation (*n* = 6). Different capital letters (A, B, C) on the top of the bars denote a significant difference (*p* < 0.05) among the NAC concentrations for the same strain, and different lowercase (a, b) letters denote a significant difference (*p* < 0.05) between the strains for the same NAC concentration. In the same way, similar capital letters on the top of the bars denote no significant difference (*p* ≥ 0.05) among the groups for the same strain, and similar lowercase letters denote no significant difference (*p* ≥ 0.05) between the strains for the same NAC concentration.

**Figure 6 microorganisms-08-00980-f006:**
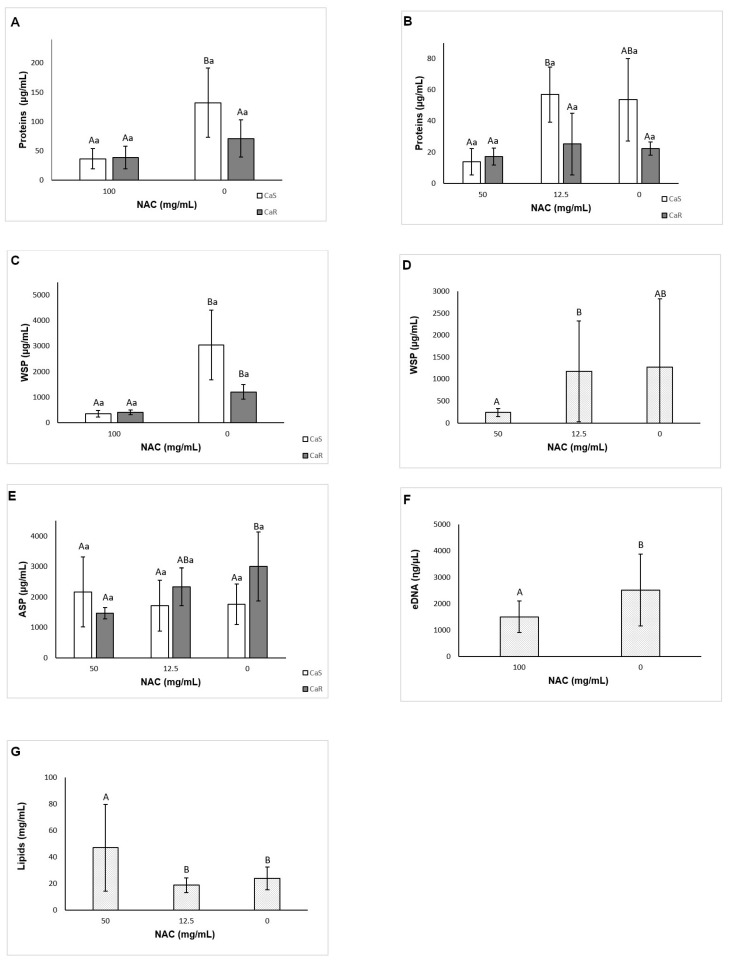
Mean values of soluble protein concentration of pre-formed biofilm (**A**) and biofilm formation (**B**); water-soluble polysaccharides (WSPs) of pre-formed biofilm (**C**) and biofilm formation (**D**); alkali-soluble polysaccharides (ASPs) from biofilm formation (**E**); eDNA of pre-formed biofilm (**F**) and lipids from biofilm formation (**G**). Error bars: standard deviation (*n* = 6). Different capital letters (A, B) on the top of the bars denote a significant difference (*p* < 0.05) among the NAC concentrations for the same strain, and different lowercase (a) letters denote a significant difference (*p* < 0.05) between the strains for the same NAC concentration. In the same way, similar capital letters on the top of the bars denote no significant difference (*p* ≥ 0.05) among the groups for the same strain, and similar lowercase letters denote no significant difference (*p* ≥ 0.05) between the strains for the same NAC concentration.

**Figure 7 microorganisms-08-00980-f007:**
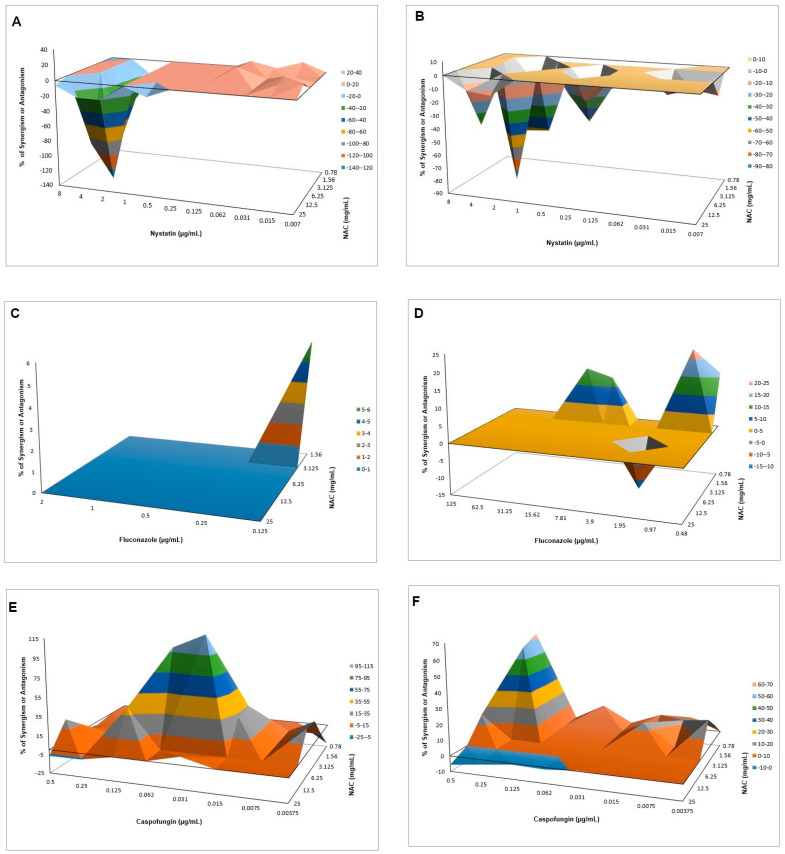
Interaction surface plot obtained from the Bliss independence analysis of the combination of NAC with nystatin (**A**,**B**), fluconazole (**C**,**D**), and caspofungin (**E**,**F**) against CaS (**A**,**C**,**E**) and CaR (**B**,**D**,**F**). Peaks above the 0 correspond to synergism and valleys below the 0 correspond to antagonism, while the 0 plane itself indicates no statistically significant interaction. Each experiment was performed in three independent occasions.

**Figure 8 microorganisms-08-00980-f008:**
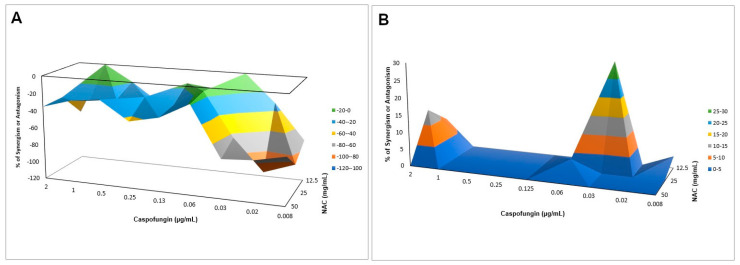
Interaction surface plot obtained from the Bliss independence analysis through the combination of NAC with caspofungin against biofilms of CaS (**A**) and CaR (**B**). Peaks above the 0 correspond to synergism and valleys below the 0 correspond to antagonism, while the 0 plane itself indicates no statistically significant interaction. Each experiment was performed in three independent occasions.

**Figure 9 microorganisms-08-00980-f009:**
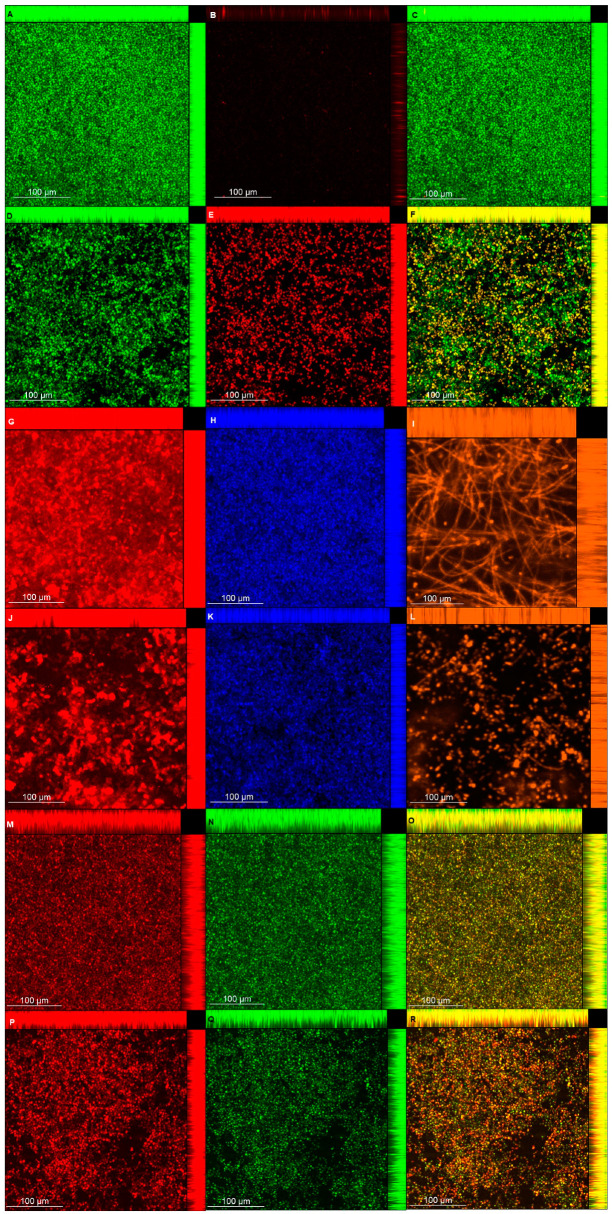
Confocal laser scanning microscopy (CLSM) images obtained for pre-formed CaS biofilm labeled with Syto9 (**A**,**D**, living cells) and propidium iodide (**B**,**E**, dead cells) for viability (**C**,**F**, overlap), without (A–C) and with NAC (**D**–**F**); concanavalin A-Alexa Fluor 594 (**G**,**J**, α-D-mannosyl and -glucosyl polysaccharides), Calcofluor White (H and K, β-glucans polysaccharide), and Sypro Ruby (**I**,**L**, proteins), without (**G**–**I**) and with NAC (**J**–**L**); and Nile Red (**M**–**R**, lipids) without (**M**–**O**) and with NAC (**P**–**R**) (**O**,**R**, overlap).

**Figure 10 microorganisms-08-00980-f010:**
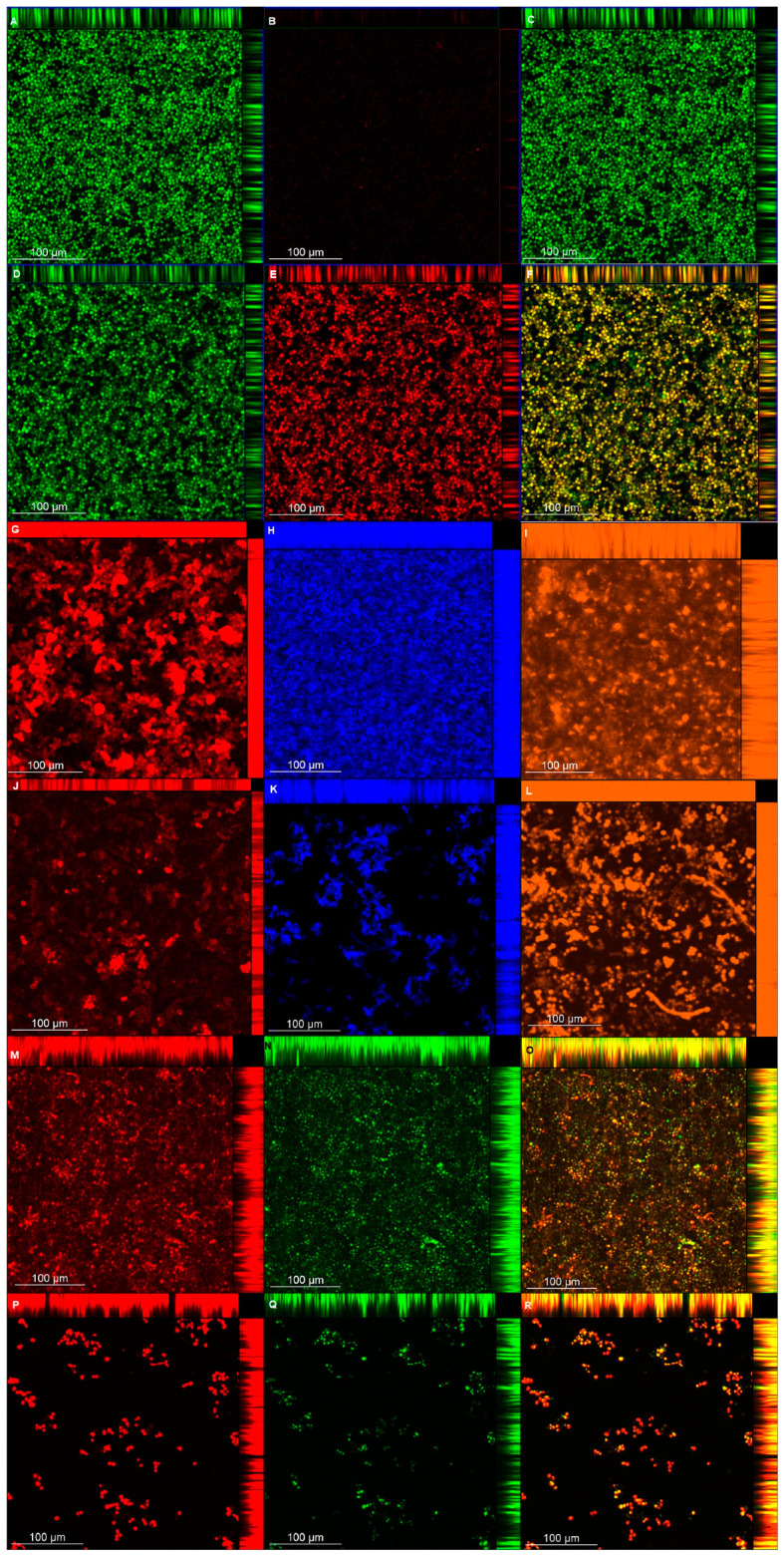
CLSM images obtained for pre-formed CaR biofilm labeled with Syto9 (**A** and **D**, living cells) and propidium iodide (**B**,**E**, dead cells) for viability (**C**,**F**, overlap), without (**A**–**C**) and with NAC (**D**–**F**); concanavalin A-Alexa Fluor 594 (**G**,**J**, α-D-mannosyl and -glucosyl polysaccharides), Calcofluor White (**H** and **K**, β-glucans polysaccharide), and Sypro Ruby (**I**,**L**, proteins), without (**G**–**I**) and with NAC (**J**–**L**); and Nile Red (**M**–**R**, lipids) without (**M**–**O**) and with NAC (**P**–**R**) (**O**,**R**, overlap).

**Table 1 microorganisms-08-00980-t001:** Concentrations of drugs used in the checkerboard assay.

Drug	CaS	CaR
NAC (mg/mL)	25, 12.5, 6.25, 3.125, 1.562, 0.781, 0	25, 12.5, 6.25, 3.125, 1.562, 0.781, 0
Nystatin (µg/mL)	8, 4, 2, 1, 0.5, 0.25, 0.125, 0.062, 0.031, 0.015, 0.007, 0	8, 4, 2, 1, 0.5, 0.25, 0.125, 0.062, 0.031, 0.015, 0.007, 0
Fluconazole (µg/mL)	2, 1, 0.5, 0.25, 0.125, 0	125, 62.5, 31.25, 15.62, 7.81, 3.9, 1.95, 0.97, 0.48, 0
Caspofungin (µg/mL)	0.5, 0.25, 0.125, 0.062, 0.031, 0.015, 0.0075, 0.0038, 0	0.5, 0.25, 0.125, 0.062, 0.031, 0.015, 0.0075, 0.0038, 0

CaS: *Candida albicans* susceptible, and CaR: *Candida albicans* resistant.

**Table 2 microorganisms-08-00980-t002:** Thickness values and fluorescence quantification (mean ± standard deviation) determined from CLSM of CaS and CaR biofilms labeled with different fluorochromes.

Fluorochromes	Thickness (µm)				Fluorescence Intensity			
	CaS		CaR		CaS		CaR	
	Contr.	NAC	Contr.	NAC	Contr.	NAC	Contr.	NAC
S9	24.00	24.00	24.00	20.00	77.68 ± 18.47	45.86 ± 5.16	74.74 ± 9.96	38.85 ± 15.64
PI					13.22 ± 5.65	28.64 ± 6.06	6.04 ± 1.03	55.06 ± 18.52
Conc	36.00	30.00	18.00	12.00	105.99 ± 15.09	89.22 ± 11.86	78.82 ± 21.11	40.83 ± 14.44
CalcW	34.38	23.04	34.38	30.56	100.33 ± 30.68	125.12 ± 27.71	178.29 ± 9.86	23.34 ± 6.91
SR	51.92	23.60	47.2	23.60	25.70 ± 7.33	49.16 ± 10.36	70.42 ± 14.40	56.25 ± 19.30
NRp	40.00	28.00	37.76	28.32	44.85 ± 15.70	37.01 ± 11.12	148.14 ± 12.66	16.51 ± 4.29
NRn					69.62 ± 21.39	74.75± 21.50	107.62 ± 8.00	15.59 ± 2.51

Fluorochrome abbreviations and respective definitions: Syto 9 (S9) for live cells; Propidium Iodide (PI) for dead cells; Concanavalin (Conc) for α-D-mannosyl and α-D-glucosyl; Calcofluor White (CalcW) for β-glucans; Sypro Ruby (SR) for proteins; Nile Red for polar lipids (NRp) and Nile Red for neutral lipids (NRn). Contr.: Control (no NAC).

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
