# Peer review of "Fungistatic Action of N-Acetylcysteine on Candida albicans Biofilms and Its Interaction with Antifungal Agents"

_microorganisms, 2020, doi:10.3390/microorganisms8070980_

Round 1

Reviewer 1 Report

The work performed and written by Nunes et al. evaluates the effect of NAC on Candida albicans biofilms with relatively good results and the combination of this mucolytic with several antifungal compounds, whose results are poor. Probably the most interesting item evaluated in this study was the inclusion of a resistant strain to Fluconazole. Obviously, nowadays the resistances against antifungal treatments represent a problem for the patients suffering from candidiasis.

Additionally, I agree with the authors that for these assays the have chosen the correct strain, a reference one, given that its ability to form biofilms is exacerbated. However, there is a problem, this is that the conclusions extracted from these assays cannot be directly translated to the clinical field, and the authors sometimes use the results from this work to make recommendations for clinicians. They have encountered that NAC concentrations under the MIC should be avoided because they increment the biofilm formation by C. albicans. The authors usually recommend to not administer NAC at “low” concentrations. What are low concentrations? Clinical isolates could respond in a different way to NAC, given that they are able to form biofilm in a low level.

Therefore, I recommend checking this kind of assimilations throughout the paper in order to reformulate them to take into account that these findings cannot be directly applied to the clinical field. Please pay attention also to the conclusions in relation with this recommendation.

Some other comments:

Abstract: The first sentence is weird, could you please rewrite it?

Line 55-56: This reference is used to argue that "the roles of each matrix component in the biofilm are still unknown", this statement could be not accurate, given that the reference is dated in 2014. From 2014 on nowadays it has had many new findings regarding to this item.

Line 158: 100 µL seems to be too small to measure the dry weight. Could you please give any set of data in order to show that they are enough for assessing the dry weight? Also, the precision weighing balance used.

Line 264: If my calculations are correct, DMSO represents 25% of the total volume, this compound should not be present in higher proportion than 10% because is usually found toxic for microorganisms. Could DMSO in that proportion modify the real effect of antifungal compounds?

Table 2: They should add statistical analysis to this table, it seems really interesting to find any difference among the controls and NAC for every comparation between strains.

Line 599: “in vitro” should be formatted in italics. Also “in vivo”

Line 601-606: There is a misunderstanding there, the reader would not know if the authors are referring to their study or to another referenced study. Pleas clarify it.

Line 620: What piece of advice would you give to clinicians? testing MIC and NAC for the isolate before NAC administration?

Line 641-644. What is a low concentration of NAC? could be assumed that every clinical isolate would respond similarly against 12.5 mg/mL? and against 100 mg/mL?

Reviewer 2 Report

The manuscript by Nunes and colleagues presents a study of N-acetylcysteine (NAC) as a fungistatic agent against biofilms formed by two strains of Candida albicans, drug-susceptible and drug-resistant. Interaction of NAC with other anti-fungal drugs was studied in the context of pathogen viability and changes in biofilm matrix composition. The authors found synergistic interaction between NAC and other antifungals. However, a cautionary note was that at lower concentrations, NAC yields the opposite effects, which makes clinical applications of such drug combinations questionable. Nevertheless, the study is of interest and reports effects of NAC on C. albicans biofilms in different strains, which was not investigated before.

I have only 2 minor comments:

The authors need to clearly indicate how many replicates were used in each experiment, either in Methods or for each Table/Figure separately. I found only one reference to triplicates but it was about only one experiment/comparison.

Captions of Figures 1,3, and 4 should be expanded to describe what all those labels (As, Aca, Ba, Ab, etc) mean.

Reviewer 3 Report

This work presents a systematic investigation of the actions of NAC toward both susceptible and resistant strains of C. albicans and corresponding biofilms. In general, the work is well organized and the claims are supported with extensive data explored in good details. However, the antifungal and antibiofilm action of NAC toward C. albicans have been reported before (Ref36, authors also mentioned in their introduction), the key point of this work is the study of the effect on resistant strains. This is, of course, important and will provide valuable information to compare both strains as studied in this work, whereas more explanations on the mechanism analysis of how the NAC works on both strains and why resistant stains respond similarly to susceptible ones will be greatly helpful to further improve the quality of this work. Apart from this point, it would be good to have the cytotoxicity profile of NAC at the dose used for antifungal use. The manuscript is well written and I would recommend acceptance of this work with minor revision considering my previous two suggestions.
